# Delay-Tolerant Algorithms for Asynchronous Distributed Online Learning

**H. Brendan McMahan**
Google, Inc.
Seattle, WA
mcmahan@google.com

**Matthew Streeter**
Duolingo, Inc.*
Pittsburgh, PA
matt@duolingo.com

## Abstract

We analyze new online gradient descent algorithms for distributed systems with large delays between gradient computations and the corresponding updates. Using insights from adaptive gradient methods, we develop algorithms that adapt not only to the sequence of gradients, but also to the precise update delays that occur. We first give an impractical algorithm that achieves a regret bound that precisely quantifies the impact of the delays. We then analyze `AdaptiveRevision`, an algorithm that is efficiently implementable and achieves comparable guarantees. The key algorithmic technique is appropriately and efficiently revising the learning rate used for previous gradient steps. Experimental results show when the delays grow large (1000 updates or more), our new algorithms perform significantly better than standard adaptive gradient methods.

## 1 Introduction

Stochastic and online gradient descent methods have proved to be extremely useful for solving large-scale machine learning problems [1, 2, 3, 4]. Recently, there has been much work on extending these algorithms to parallel and distributed systems [5, 6, 7, 8, 9]. In particular, Recht et al. [10] and Duchi et al. [11] have shown that standard stochastic algorithms essentially "work" even when updates are applied asynchronously by many threads. Our experiments confirm this for moderate amounts of parallelism (say 100 threads), but show that for large amounts of parallelism (as in a distributed system, with say 1000 threads spread over many machines), performance can degrade significantly. To address this, we develop new algorithms that adapt to both the data and the amount of parallelism.

Adaptive gradient (AdaGrad) methods [12, 13] have proved remarkably effective for real-world problems, particularly on sparse data (for example, text classification with bag-of-words features). The key idea behind these algorithms is to prove a general regret bound in terms of an arbitrary sequence of non-increasing learning rates and the full sequence of gradients, and then to define an adaptive method for choosing the learning rates as a function of the gradients seen so far, so as to minimize the final bound when the learning rates are plugged in. We extend this idea to the parallel setting, by developing a general regret bound that depends on both the gradients and the exact update delays that occur (rather than say an upper bound on delays). We then present `AdaptiveRevision`, an algorithm for choosing learning rates and efficiently revising past learning-rate choices that strives to minimize this bound. In addition to providing an adaptive regret bound (which recovers the standard AdaGrad bound in the case of no delays), we demonstrate excellent empirical performance.

**Problem Setting and Notation** We consider a computation model where one or more computation units (a thread in a parallel implementation or a full machine in a distributed system) store and

update the model $x \in \mathbb{R}^n$, and another larger set of computation units perform feature extraction and prediction. We call the first type the `Updaters` (since they apply the gradient updates) and the second type the `Readers` (since they read coefficients stored by the `Updaters`). Because the `Readers` and `Updaters` may reside on different machines, perhaps located in different parts of the world, communication between them is not instantaneous. Thus, when making a prediction, a `Reader` will generally be using a coefficient vector that is somewhat stale relative to the most recent version being served by the `Updaters`.

As one application of this model, consider the problem of predicting click-through rates for sponsored search ads using a generalized linear model [14, 15]. While the coefficient vector may be stored and updated centrally, predictions must be available in milliseconds in any part of the world. This leads naturally to an architecture in which a large number of `Readers` maintain local copies of the coefficient vector, sending updates to the `Updaters` and periodically requesting fresh coefficients from them. As another application, this model encompasses the Parameter Server/ Model Replica split of Downpour SGD [16].

Our bounds apply to general online convex optimization [4], which encompasses the problem of predicting with a generalized linear model (models where the prediction is a function of $a_t \cdot x_t$, where $a_t$ is a feature vector and $x_t$ are model coefficients). We analyze the algorithm on a sequence of $\tau = 1, ..., T$ rounds; for the moment, we index rounds based on when each prediction is made. On each round, a convex loss function $f_\tau$ arrives at a `Reader`, the `Reader` predicts with $x_\tau \in \mathbb{R}^n$ and incurs loss $f_\tau(x_\tau)$. The `Reader` then computes a subgradient $g_\tau \in \partial f_\tau(x_\tau)$. For each coordinate $i$ where $g_{\tau,i}$ is nonzero, the `Reader` sends an update to the `Updater`(s) for those coefficients. We are particularly concerned with sparse data, where $n$ is very large, say $10^6 - 10^9$, but any particular training example has only a small fraction of the features $a_{t,i}$ that take non-zero values.

The regret against a comparator $x^* \in \mathbb{R}^n$ is

$$\text{Regret}(x^*) \equiv \sum_{\tau=1}^{T} f_\tau(x_\tau) - f_\tau(x^*). \tag{1}$$

Our primary theoretical contributions are upper bounds on the regret of our algorithms.

We assume a fully asynchronous model, where the delays in the read requests and update requests can be different for different coefficients even for the same training event. This leads to a combinatorial explosion in potential interleavings of these operations, making fine-grained adaptive analysis quite difficult. Our primary technique for addressing this will be the linearization of loss functions, a standard tool in online convex optimization which takes on increased importance in the parallel setting. An immediate consequence of convexity is that given a general convex loss function $f_\tau$, with $g_\tau \in \partial f_\tau(x_\tau)$, for any $x^*$, we have $f_\tau(x_\tau) - f_\tau(x^*) \le g_\tau \cdot (x_\tau - x^*)$. One of the key observations of Zinkevich [1] is that by plugging this inequality into (1), we see that if we can guarantee low regret against linear functions, we can provide the same guarantees against arbitrary convex functions. Further, expanding the dot products and re-arranging the sum, we can write

$$\text{Regret}(x^*) \equiv \sum_{i=1}^{n} \text{Regret}_i(x_i^*) \qquad \text{where} \qquad \text{Regret}_i(x_i^*) = \sum_{\tau=1}^{T} g_{\tau,i}(x_{\tau,i} - x_i^*). \tag{2}$$

If we consider algorithms where the updates are also coordinate decomposable (that is, the update to coordinate $i$ can be applied independently of the update of coordinate $j$), then we can bound $\text{Regret}(x^*)$ by proving a per-coordinate bound for linear functions and then summing across coordinates. In fact, our computation architecture already assumes a coordinate decomposable algorithm since this lets us avoid synchronizing the `Updates`, and so in addition to leading to more efficient algorithms, this approach will greatly simplify the analysis. The proofs of Duchi et al. [11] take a similar approach.

**Bounding per-coordinate regret** Given the above, we will design and analyze asynchronous one-dimensional algorithms which can be run independently on each coordinate of the true learning problem. For each coordinate, each `Read` and `Update` is assumed to be an atomic operation. It will be critical to adopt an indexing scheme different than the prediction-based indexing $\tau$ used above. The net result will be bounding the sum of (2), but we will actually re-order the sum to make the analysis easier. Critically, this ordering could be *different* for different coordinates, and

so considering one coordinate at a time simplifies the analysis considerably.[1] We index time by the order of the Updates, so the index $t$ is such that $g_t$ is the gradient associated with the $t$th update applied and $x_t$ is the value of the coefficient immediately *before* the update for $g_t$ is applied. Then, the Online Gradient Descent (OGD) update consists of exactly the assumed-atomic operation

$$x_{t+1} = x_t - \eta_t g_t, \tag{3}$$

where $\eta_t$ is a learning-rate. Let $r(t) \in \{1, \dots, t\}$ be the index such that $x_{r(t)}$ was the value of the coefficient used by the Reader to compute $g_t$ (and to predict on the corresponding example). That is, update $r(t) - 1$ completed before the Read for $g_t$, but update $r(t)$ completed after. Thus, our loss (for coordinate $i$) is $g_t x_{r(t)}$, and we desire a bound on

$$\text{Regret}_i(x^*) = \sum_{t=1}^{T} g_t(x_{r(t)} - x^*).$$

**Main result and related work** We say an update $s$ is outstanding at time $t$ if the Read for Update $s$ occurs before update $t$, but the Update occurs after: precisely, $s$ is outstanding at $t$ if $r(s) \leq t < s$. We let $\mathcal{F}_t \equiv \{s \mid r(s) \leq t < s\}$ be the set of updates outstanding at time $t$. We call the sum of these gradients the *forward gradient sum*, $g_t^{\text{fwd}} \equiv \sum_{s \in \mathcal{F}_t} g_s$. Then, ignoring constant factors and terms independent of $T$, we show that AdaptiveRevision has a per-coordinate bound of the form

$$\text{Regret} \leq \sqrt{\sum_{t=1}^{T} g_t^2 + g_t g_t^{\text{fwd}}}. \tag{4}$$

Theorem 3 gives the precise result as well as the $n$-dimensional version. Observe that without any delays, $g_t^{\text{fwd}} = 0$, and we arrive at the standard AdaGrad-style bound. To prove the bound for AdaptiveRevision, we require an additional InOrder assumption on the delays, namely that for any indexes $s_1$ and $s_2$, if $r(s_1) < r(s_2)$ then $s_1 < s_2$. This assumption should be approximately satisfied most of the time for realistic delay distributions, and even under a more pathological delay distributions (delays uniform on $\{0, \dots, m\}$ rather than more tightly grouped around a mean delay), our experiments show excellent performance for AdaptiveRevision.

The key challenge is that unlike in the AdaGrad case, conceptually we need to know gradients that have not yet been computed in order to calculate the optimal learning rate. We surmount this by using an algorithm that not only chooses learning rates adaptively, but also *revises* previous gradient steps. Critically, these revisions require only moderate additional storage and network cost: we store a sum of gradients along with each coefficient, and for each Read, we remember the value of this gradient sum at the time of the Read until the corresponding Update occurs. This later storage can essentially be implemented on the network, if the gradient sum is sent from the Updater to the Reader and back again, ensuring it is available exactly when needed. This is the approach taken in the pseudocode of Algorithm 1.

Against a true adversary and a maximum delay of $m$, in general we cannot do better than just training synchronously on a single machine using a $1/m$ fraction of the data. Our results surmount this issue by producing strongly data-dependent bounds: we do not expect fully adversarial gradients and delays in practice, and so on real data the bound we prove still gives interesting results. In fact, we can essentially recover the guarantees for AsyncAdaGrad from Duchi et al. [11], which rely on stochastic assumptions on the sparsity of the data, by applying the same assumptions to our bound. To simplify the comparison, WLOG we consider a 1-dimensional problem where $\|x^*\|_2 = 1$, $\|g_t\|_2 \leq 1$, and we have the stochastic assumption that each $g_t$ is exactly 0 independently with probability $p$ (implying $M_j = 1$, $M = 1$, and $\mathsf{M}^2 = p$ in their notation). Then, simple calculations (given in Appendix B) show our bound for AdaptiveRevision implies a bound on expected regret of $\mathcal{O}\big(\sqrt{(1 + mp)pT}\big)$ without knowledge of $p$ or $m$, ignoring terms independent of $T$.[2] AsyncAdaGrad achieves the same bound, but critically this requires knowledge of both $p$ and

$m$ in advance in order to tune the learning rate appropriately (in the general $n$-dimensional case, this would mean knowing not just one parameter $p$, but a separate sparsity parameter $p_j$ for each coordinate, and then using an appropriate per-coordinate scaling of the learning rate depending on this); without such knowledge, AsyncAdaGrad only obtains the much worse bound $\mathcal{O}\big((1 + mp)\sqrt{pT}\big)$. `AdaptiveRevision` will also provide significantly better guarantees if most of the delays are much less than the maximum, or if the data is only approximately sparse (e.g., many $g_t = 10^{-6}$ rather than exactly 0). The above analysis also makes a worst-case assumption on the $g_t g_t^{\mathrm{fwd}}$ terms, but in practice many gradients in $g_t^{\mathrm{fwd}}$ are likely to have opposite signs and cancel out, a fact our algorithm and bounds can exploit.

## 2 Algorithms and Analysis

We first introduce some additional definitions. Let $o(t) \equiv \max \mathcal{F}_t \cup \{t\}$, the index of the highest update outstanding at time $t$, or $t$ itself if nothing is outstanding. The sets $\mathcal{F}_t$ fully specify the delay pattern. In light of (4), we further define $G_t^{\mathrm{fwd}} \equiv g_t^2 + 2g_t g_t^{\mathrm{fwd}}$. We also define $\mathcal{B}_t$, the set of updates applied while update $t$ was outstanding. Under our notation, this set is easily defined as $\mathcal{B}_t = \{r(t), \dots, t - 1\}$ (or the empty set if $r(t) = t$, so in particular $\mathcal{B}_1 = \emptyset$). We will also frequently use the *backward gradient sum*, $g_t^{\mathrm{bck}} \equiv \sum_{s=r(t)}^{t-1} g_s$. These vectors most often appear in the products $G_t^{\mathrm{bck}} \equiv g_t^2 + 2g_t g_t^{\mathrm{bck}}$. Figure 3 in Appendix A shows a variety of delay patterns and gives a visual representation of the sums $G^{\mathrm{fwd}}$ and $G^{\mathrm{bck}}$. We say the delay is (upper) bounded by $m$ if $t - r(t) \le m$ for all $t$, which implies $|\mathcal{F}_t| \le m$ and $|\mathcal{B}_t| \le m$. Note that if $m = 0$ then $r(t) = t$. We use the compressed summation notation $c_{1:t} \equiv \sum_{s=1}^{t} c_s$ for vectors, scalars, and functions.

Our analysis builds on the following simple but fundamental result (Appendix C contains all proofs and lemmas omitted here).

**Lemma 1.** *Given any non-increasing learning-rate schedule $\eta_t$, define $\sigma_t$ where $\sigma_1 = 1/\eta_1$ and $\sigma_t = 1/\eta_t - 1/\eta_{t-1}$ for $t > 1$, so $\eta_t = 1/\sigma_{1:t}$. Then, for any delay schedule, unprojected online gradient descent achieves, for any $x^* \in \mathbb{R}$,*

$$\mathrm{Regret}(x^*) \le \frac{(2R_T)^2}{2\eta_T} + \frac{1}{2}\sum_{t=1}^{T} \eta_t G_t^{\mathrm{fwd}} \qquad where \qquad (2R_T)^2 \equiv \sum_{t=1}^{T} \frac{\sigma_t}{\sigma_{1:T}} |x^* - x_t|^2.$$

*Proof.* Given how we have indexed time, we can consider the regret of a hypothetical online gradient descent algorithm that plays $x_t$ and then observes $g_t$, since this corresponds exactly to the update (3). We can then bound regret for this hypothetical setting using a simple modification to standard bound for OGD [1],

$$\sum_{t=1}^{T} g_t \cdot x_t - g_{1:T} \cdot x^* \le \sum_{t=1}^{T} \frac{\sigma_t}{2} |x^* - x_t|^2 + \frac{1}{2}\sum_{t=1}^{T} \eta_t g_t^2.$$

The actual algorithm used $x_{r(t)}$ to predict on $g_t$, not $x_t$, so we can bound its Regret by

$$\mathrm{Regret} \le \frac{(2R_T)^2}{2\eta_T} + \frac{1}{2}\sum_{t=1}^{T} \eta_t g_t^2 + \sum_{t=1}^{T} g_t(x_{r(t)} - x_t). \tag{5}$$

Recalling $x_{t+1} = x_t - \eta_t g_t$, observe that $x_{r(t)} - x_t = \sum_{s=r(t)}^{t-1} \eta_s g_s, = \sum_{s \in \mathcal{B}_t} \eta_s g_s$ and so

$$\sum_{t=1}^{T} g_t(x_{r(t)} - x_t) = \sum_{t=1}^{T} g_t \sum_{s \in \mathcal{B}_t} \eta_s g_s = \sum_{s=1}^{T} \eta_s g_s \sum_{t \in \mathcal{F}_s} g_t = \sum_{s=1}^{T} \eta_s g_s g_s^{\mathrm{fwd}},$$

using Lemma 4(E) from the Appendix to re-order the sum. Plugging into (5) completes the proof. $\qquad\square$

For projected online gradient descent, by projecting onto a feasible set of radius $R$ and assuming $x^*$ is in this set, we immediately get $|x^* - x_t| \le 2R$. Without projecting, we get a more adaptive bound which depends on the weighted quadratic mean $2R_T$. Though less standard, we choose to

analyze the unprojected variant of the algorithm for two reasons. First, our analysis rests heavily on the ability to represent points played by our algorithms exactly as weighted sums of past gradients, a property not preserved when projection is invoked. More importantly, we know of no experiments on real-world prediction problems (where any $x \in \mathbb{R}^n$ is a valid model) where the projected algorithm actually performs better. In our experience, once the learning-rate schedule is tuned appropriately, the resulting $R_T$ values will not be more than a constant factor of $\|x^*\|$. This makes intuitive sense in the stochastic case, where it is known that averages of the $x_t$ should in fact converge to $x^*$.[3] For learning rate tuning we assume we know in advance a constant $\tilde{R}$ such that $R_T \leq \tilde{R}$; again, in practice this is roughly equivalent to assuming we know $\|x^*\|$ in advance in order to choose the feasible set.

Our first algorithm, `HypFwd` (for Hypothetical-Forward), assumes it has knowledge of all the gradients, so it can optimize its learning rates to minimize the above bound. If there are no delays, that is, $g_t^{\text{fwd}} = 0$ for all $t$, then this immediately gives rise to a standard AdaGrad-style online gradient descent method. If there are delays, the $G_t^{\text{fwd}}$ terms could be large, implying the optimal learning rates should be smaller. Unfortunately, it is impossible for a real algorithm to know $g_t^{\text{fwd}}$ when $\eta_t$ is chosen. To work toward a practical algorithm, we introduce `HypBack`, which achieves similar guarantees (but is still impractical). Finally, we introduce `AdaptiveRevision`, which plays points very similar to `HypBack`, but can be implemented efficiently. Since we will need non-increasing learning rates, it will be useful to define $\tilde{G}_{1:t}^{\text{bck}} \equiv \max_{s \leq t} G_{1:s}^{\text{bck}}$ and $\tilde{G}_{1:t}^{\text{fwd}} \equiv \max_{s \leq t} G_{1:s}^{\text{fwd}}$. In practice, we expect $\tilde{G}_{1:T}^{\text{bck}}$ to be close to $G_{1:T}^{\text{bck}}$. We assume WLOG that $G_1^{\text{fwd}} > 0$, which at worst adds a negligible additive constant to our regret.

**Algorithm `HypFwd`** This algorithm "cheats" by using the forward sum $g_t^{\text{fwd}}$ to choose $\eta_t$,

$$\eta_t = \frac{\alpha}{\sqrt{\tilde{G}_{1:t}^{\text{fwd}}}} \tag{6}$$

for an appropriate scaling parameter $\alpha > 0$. Then, Lemma 1 combined with the technical inequality of Corollary 10 (given in Appendix D) gives

$$\text{Regret} \leq 2\sqrt{2}\tilde{R}\sqrt{\tilde{G}_{1:T}^{\text{fwd}}}. \tag{7}$$

when we take $\alpha = \sqrt{2}\tilde{R}$ (recalling $\tilde{R} \geq R_T$). If there are no delays, this bound reduces to the standard bound $2\sqrt{2}\tilde{R}\sqrt{\sum_{t=1}^{T} g_t^2}$. With delays, however, this is a hypothetical algorithm, because it is generally not possible to know $g_t^{\text{fwd}}$ when update $t$ is applied. However, we can implement this algorithm efficiently in a single-machine simulation, and it performs very well (see Section 3). Thus, our goal is to find an efficiently implementable algorithm that achieves comparable results in practice and also matches this regret bound.

**Algorithm `HypBack`** The next step in the analysis is to show that a second hypothetical algorithm, `HypBack`, approximates the regret bound of (7). This algorithm plays

$$\hat{x}_{t+1} = -\sum_{s=1}^{t} \hat{\eta}_s g_s \qquad \text{where} \qquad \hat{\eta}_t = \frac{\alpha}{\sqrt{\tilde{G}_{1:o(t)}^{\text{bck}} + G_0}} \tag{8}$$

is a learning rate with parameters $\alpha$ and $G_0$. This is a hypothetical algorithm, since we also can't (efficiently) know $G_{1:o(t)}^{\text{bck}}$ on round $t$. We prove the following guarantee:

**Lemma 2.** *Suppose delays bounded by $m$ and $|g_t| \leq L$. Then when the `InOrder` property holds, `HypBack` with $\alpha = \sqrt{2}\tilde{R}$ and $G_0 = m^2L^2$ has*

$$\text{Regret} \leq 2\sqrt{2}\tilde{R}\sqrt{\tilde{G}_{1:T}^{\text{fwd}}} + 2\tilde{R}mL.$$

**Algorithm 1** Algorithm `AdaptiveRevision`

---

**Procedure `Read`(loss function $f$):**
    Read $(x_i, \bar{g}_i)$ from the `Updaters` for all necessary coordinates
    Calculate a subgradient $g \in \partial f(x)$
    **for** each coordinate $i$ with a non-zero gradient **do**
        Send an update tuple $(g \leftarrow g_i, \bar{g}^{\text{old}} \leftarrow \bar{g}_i)$ to the `Updater` for coordinate $i$

**Procedure `Update`$(g, \bar{g}^{\text{old}})$:**   *The Updater initializes state $(\bar{g} \leftarrow 0, z \leftarrow 1, z' \leftarrow 1, x \leftarrow 0)$ per coordinate.*
    *Do the following atomically:*

| | | |
|---|---|---|
| $g^{\text{bck}} \leftarrow \bar{g} - \bar{g}^{\text{old}}$ | | *For analysis, assign index $t$ to the current update.* |
| $\eta^{\text{old}} \leftarrow \frac{\alpha}{\sqrt{z'}}$ | | *Invariant: effective $\eta$ for all $g^{\text{bck}}$.* |
| $z \quad \leftarrow z + g^2 + 2g \cdot g^{\text{bck}}; z' \leftarrow \max(z, z')$ | | *Maintain $z = G^{\text{bck}}_{1:t}$ and $z' = \tilde{G}^{\text{bck}}_{1:t}$, to enforce non-increasing $\eta$.* |
| $\eta \quad \leftarrow \frac{\alpha}{\sqrt{z'}}$ | | *New learning rate.* |
| $x \quad \leftarrow x - \eta g$ | | *The main gradient-descent update.* |
| $x \quad \leftarrow x + (\eta^{\text{old}} - \eta)g^{\text{bck}}$ | | ***Apply adaptive revision of some previous steps.*** |
| $\bar{g} \quad \leftarrow \bar{g} + g$ | | *Maintain $\bar{g} = g_{1:t}$.* |

---

**Algorithm `AdaptiveRevision`**   Now that we have shown that `HypBack` is effective, we can describe `AdaptiveRevision`, which efficiently approximates `HypBack`. We then analyze this new algorithm by showing its loss is close to the loss of `HypBack`. Pseudo-code for the algorithm as implemented for the experiments is given in Algorithm 1; we now give an equivalent expression for the algorithm under the `InOrder` assumption. Let $\beta_t$ be the learning rate based on $\tilde{G}^{\text{bck}}_{1:t}$, $\beta_t = \alpha/\sqrt{\tilde{G}^{\text{bck}}_{1:t} + G_0}$. Then, `AdaptiveRevision` plays the points

$$x_{t+1} = \sum_{s=1}^{t} \eta_s^t g_s \qquad \text{where} \qquad \eta_s^t = \beta_{\min(t, o(s))}. \tag{9}$$

When $s \ll t$ then we will usually have $\min(t, o(s)) = o(s)$, and so we see that $\eta_s^t = \beta_{o(s)} = \hat{\eta}_s$, and so the effective learning rate applied to gradient $g_s$ is the same one `HypBack` would have used (namely $\hat{\eta}_s$); thus, the only difference between `AdaptiveRevision` and `HypBack` is on the leading edge, where $o(s) > t$. See Figure 4 in Appendix A for an example. When `InOrder` holds, Lemma 6 (in Appendix C) shows Algorithm 1 plays the points specified by (9).

Given Lemma 2, it is sufficient to show that the difference between the loss of `HypBack` and the loss of `AdaptiveRevision` is small. Lemma 8 (in the appendix) accomplishes this, showing that under the `InOrder` assumption and with $G_0 = m^2 L^2$ the difference in loss is at most $2\alpha Lm$ (a quantity independent of $T$). Our main theorem is then a direct consequence of Lemma 2 and Lemma 8:

**Theorem 3.** *Under an `InOrder` delay pattern with a maximum delay of at most $m$, the `AdaptiveRevision` algorithm guarantees* Regret $\leq 2\sqrt{2}\tilde{R}\sqrt{\tilde{G}^{\text{fwd}}_{1:T}} + (2\sqrt{2} + 2)\tilde{R}mL$ *when we take $G_0 = m^2 L^2$ and $\alpha = \sqrt{2}\tilde{R}$. Applied on a per-coordinate basis to an $n$-dimensional problem, we have*

$$\text{Regret} \leq 2\sqrt{2}\tilde{R} \sum_{i=1}^{n} \sqrt{\sum_{t=1}^{T} \left( g_{t,i}^2 + 2 \sum_{s \in \mathcal{F}_{t,i}} g_{s,i} g_{s,i} \right)} + n(2\sqrt{2} + 2)\tilde{R}mL.$$

We note the $n$-dimensional guarantee is at most $\mathcal{O}(n\tilde{R}L\sqrt{Tm})$, which matches the lower bound for the feasible set $[-R, R]^n$ and $g_t \in [-L, L]^n$ up to the difference between $\tilde{R}$ and $R$ (see, for example, Langford et al. [18]).[4] Our point, of course, is that for real data our bound will often be much much better.

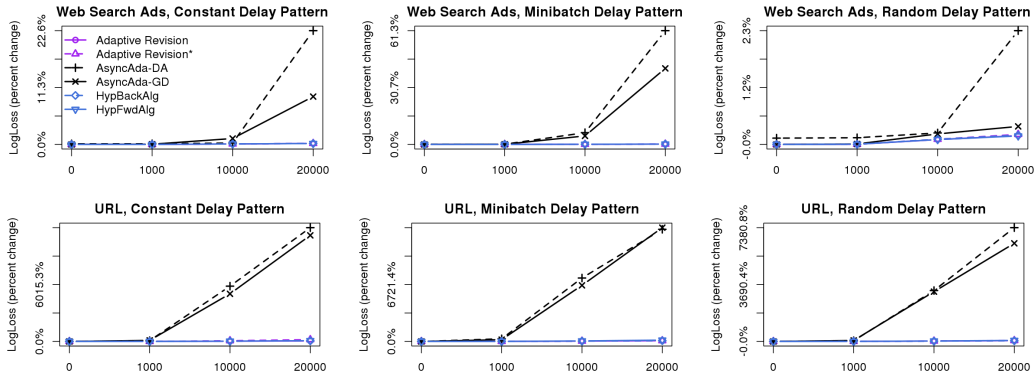

Figure 1: Accuracy as a function of update delays, with learning rate scale factors optimized for each algorithm and dataset for the zero delay case. The $x$-axis is non-linear. The results are qualitatively similar across the plots, but note the differences in the $y$-axis ranges. In particular, the random delay pattern appears to hurt performance significantly less than either the minibatch or constant delay patterns.

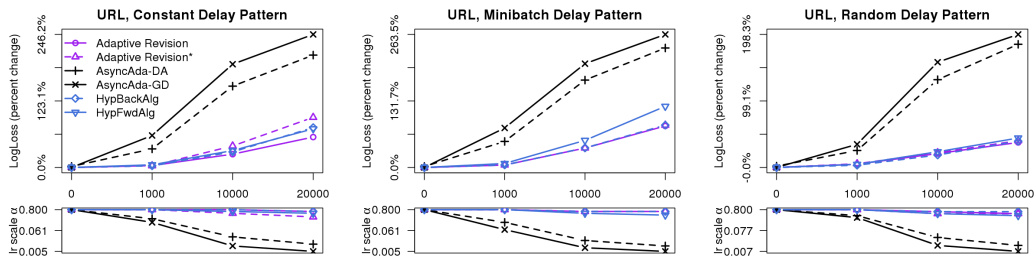

Figure 2: Accuracy as a function of update delays, with learning rate scale factors optimized as a function of the delay. The lower plot in each group shows the best learning rate scale $\alpha$ on a log-scale.

## 3   Experiments

We study the performance of both hypothetical algorithms and `AdaptiveRevision` on two real-world medium-sized datasets. We simulate the update delays using an update queue, which allows us to implement the hypothetical algorithms and also lets us precisely control both the exact delays as well as the delay pattern. We compare to the dual-averaging AsyncAdaGrad algorithm of Duchi et al. [11] (AsyncAda-DA in the figures), as well as asynchronous AdaGrad gradient descent (AsyncAda-GD), which can be thought of as `AdaptiveRevision` with all $g^{\text{bck}}$ set to zero and no revision step. As analyzed, `AdaptiveRevision` stores an extra variable ($z'$) in order to enforce a non-increasing learning rate. In practice, we found this had a negligible impact; in the plots above, `AdaptiveRevision`* denotes the algorithm without this check. With this improvement `AdaptiveRevision` stores three numbers per coefficient, versus the two stored by AsyncAda-grad DA or GD.

We consider three different delay patterns, which we parameterize by $D$, the *average* delay; this yields a more fair comparison across the delay patterns than using the the maximum delay $m$. We consider: 1) constant delays, where all updates (except at the beginning and the end of the dataset) have a delay of exactly $D$ (e.g., rows (B) and (C) in Figure 3 in the Appendix); 2) A minibatch delay pattern[5], where $2D+1$ `Reads` occur, followed by $2D+1$ `Updates`; and 3) a random delay pattern, where the delays are chosen uniformly from the set $\{0, \dots, 2D\}$, so again the mean delay is $D$. The first two patterns satisfy `InOrder`, but the third does not.

We evaluate on two datasets. The first is a web search advertising dataset from a large search engine. The dataset consists of about $3.1 \times 10^6$ training examples with a large number of sparse anonymized features based on the ad and query text. Each example is labeled $\{-1, 1\}$ based on whether or not the person doing the query clicked on the ad. The second is a shuffled version of the malicious URL dataset as described by Ma et al. [19] ($2.4 \times 10^6$ examples, $3.2 \times 10^6$ features).[6] For each of these datasets we trained a logistic regression model, and evaluated using the logistic loss (LogLoss). That is, for an example with feature vector $a \in \mathbb{R}^n$ and label $y \in \{-1, 1\}$, the loss is given by $\ell(x, (a, y)) = \log(1 + \exp(-y\, a \cdot x))$. Following the spirit of our regret bounds, we evaluate the models online, making a single pass over the data and computing accuracy metrics on the predictions made by the model immediately before it trained on each example (i.e., progressive validation). To avoid possible transient behavior, we only report metrics for the predictions on the second half of each dataset, though this choice does not change the results significantly.

The exact parametrization of the learning rate schedule is particularly important with delayed updates. We follow the common practice of taking learning rates of the form $\eta_t = \alpha/\sqrt{S_t + 1}$, where $S_t$ is the appropriate learning rate statistic for the given algorithm, e.g., $\tilde{G}^{\text{bck}}_{1:o(t)}$ for `HypBack` or $\sum_{s=1}^t g_s^2$ for vanilla AdaGrad. In the analysis, we use $G_0 = m^2 L^2$ rather than $G_0 = 1$; we believe $G_0 = 1$ will generally be a better choice in practice, though we did not optimize this choice.[7] When we optimize $\alpha$, we choose the best setting from a grid $\{\alpha_0(1.25)^i \mid i \in \mathbb{N}\}$, where $\alpha_0$ is an initial guess for each dataset.

All figures give the average delay $D$ on the $x$-axis. For Figure 1, for each dataset and algorithm, we optimized $\alpha$ in the zero delay ($D = m = 0$) case, and fixed this parameter as the average delay $D$ increases. This leads to very bad performance for standard AdaGrad DA and GD as $D$ gets large. In Figure 2, we optimized $\alpha$ individually for each delay level; we plot the accuracy as before, with the lower plot showing the optimal learning rate scaling $\alpha$ on a log-scale. The optimal learning rate scaling for GD and DA decrease by *two orders of magnitude* as the delays increase. However, even with this tuning they do not obtain the performance of `AdaptiveRevision`. The performance of `AdaptiveRevision` (and `HypBack` and `HypFwd`) is slightly improved by lowering the learning rate as delays increase, but the effect is comparatively very minor. As anticipated, the performance for `AdaptiveRevision`, `HypBack`, and `HypFwd` are closely grouped.

`AdaptiveRevision`'s delay tolerance can lead to enormous speedups in practice. For example, the leftmost plot of Figure 2 shows that `AdaptiveRevision` achieves better accuracy with an update delay of 10,000 than AsyncAda-DA achieves with a delay of 1000. Because update delays are proportional to the number of `Readers`, this means that `AdaptiveRevision` can be used to train a model an order of magnitude faster than AsyncAda-DA, with no reduction in accuracy. This allows for much faster iteration when data sets are large and parallelism is cheap, which is the case in important real-world problems such as ad click-through rate prediction [14].

## 4    Conclusions and Future Work

We have demonstrated that adaptive tuning and revision of per-coordinate learning rates for distributed gradient descent can significantly improve accuracy as the update delays become large. The key algorithmic technique is maintaining a sum of gradients, which allows the adjustment of all learning rates for gradient updates that occurred between the current `Update` and its `Read`. The analysis method is novel, but is also somewhat indirect; an interesting open question is finding a general analysis framework for algorithms of this style. Ideally such an analysis would also remove the technical need for the `InOrder` assumption, and also allow for the analysis of `AdaptiveRevision` variants of OGD with Projection and Dual Averaging.

## Footnotes

*Work performed while at Google, Inc.

[1] Our analysis could be extended to non-coordinate-decomposable algorithms, but then the full gradient update across all coordinates would need to be atomic. This case is less interesting due to the computational overhead.

[2] In the analysis, we choose the parameter $G_0$ based on an upper bound $m$ on the delay, but this only impacts an additive term independent of $T$.

[3]For example, the arguments of Nemirovski et al. [17, Sec 2.2] hold for unprojected gradient descent.

[4]To compare to regret bounds stated in terms of $L_2$ bounds on the feasible set and the gradients, note for $g_t \in [-L, L]^n$ we have $\|g_t\|_2 \leq \sqrt{n}L$, and similarly for $x \in [-R, R]^n$ we have $\|x\|_2 \leq \sqrt{n}R$, so the dependence on $n$ is a necessary consequence of using these norms, which are quite natural for sparse problems.

[5]It is straightforward to show that under this delay pattern, when we do not enforcing non-increasing learning rates, `AdaptiveRevision` and `HypBack` are in fact equivalent to standard AdaGrad run on the mini-batches (that is, with one update per minibatch using the combined minibatch gradient sum).

[6]We also ran experiments on the `rcv1.binary` training dataset ($0.6 \times 10^6$ examples, $0.05 \times 10^6$ features) from Chang and Lin [20]; results were qualitatively very similar to those for the URL dataset.

[7]The main purpose of choosing a larger $G_0$ in the theorems was to make the performance of `HypBack` and `AdaptiveRevision` provably close to that of `HypFwd`, even in the worst case. On real data, the performance of the algorithms will typically be close even with $G_0 = 1$.

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
