[Supplementary Material]

# A  Delay Pattern Examples

**(A) Minibatches of size 3**

Left matrix (columns $g_1\,g_2\,g_3\,g_4\,g_5\,g_6$):

| | $g_1$ | $g_2$ | $g_3$ | $g_4$ | $g_5$ | $g_6$ |
|---|---|---|---|---|---|---|
| $g_1$ | 1 | 2 | 3 | | | |
| $g_2$ | 2 | 2 | 3 | | | |
| $g_3$ | 3 | 3 | 3 | | | |
| $g_4$ | | | | 4 | 5 | 6 |
| $g_5$ | | | | 5 | 5 | 6 |
| $g_6$ | | | | 6 | 6 | 6 |

Right matrix:

| | $g_1$ | $g_2$ | $g_3$ | $g_4$ | $g_5$ | $g_6$ |
|---|---|---|---|---|---|---|
| $g_1$ | 1 | 1 | 1 | | | |
| $g_2$ | 1 | 2 | 2 | | | |
| $g_3$ | 1 | 2 | 3 | | | |
| $g_4$ | | | | 4 | 4 | 4 |
| $g_5$ | | | | 4 | 5 | 5 |
| $g_6$ | | | | 4 | 5 | 6 |

| | | |
|---|---|---|
| Read 1,2,3 | Update 1 | $\mathcal{F}_1 = \{2,3\}$ |
| | Update 2 | $\mathcal{F}_2 = \{3\}$ |
| | Update 3 | $\mathcal{F}_3 = \{\}$ |
| Read 4,5,6 | Update 4 | $\mathcal{F}_4 = \{5,6\}$ |
| | Update 5 | $\mathcal{F}_5 = \{6\}$ |
| | Update 6 | $\mathcal{F}_6 = \{\}$ |

**(B) Fixed delay of 1**

Left matrix:

| | $g_1$ | $g_2$ | $g_3$ | $g_4$ | $g_5$ | $g_6$ |
|---|---|---|---|---|---|---|
| $g_1$ | 1 | 2 | | | | |
| $g_2$ | 2 | 2 | 3 | | | |
| $g_3$ | | 3 | 3 | 4 | | |
| $g_4$ | | | 4 | 4 | 5 | |
| $g_5$ | | | | 5 | 5 | 6 |
| $g_6$ | | | | | 6 | 6 |

Right matrix:

| | $g_1$ | $g_2$ | $g_3$ | $g_4$ | $g_5$ | $g_6$ |
|---|---|---|---|---|---|---|
| $g_1$ | 1 | 1 | | | | |
| $g_2$ | 1 | 2 | 2 | | | |
| $g_3$ | | 2 | 3 | 3 | | |
| $g_4$ | | | 3 | 4 | 4 | |
| $g_5$ | | | | 4 | 5 | 5 |
| $g_6$ | | | | | 5 | 6 |

| | | |
|---|---|---|
| Read 1, 2 | Update 1 | $\mathcal{F}_1 = \{2\}$ |
| Read 3 | Update 2 | $\mathcal{F}_2 = \{3\}$ |
| Read 4 | Update 3 | $\mathcal{F}_3 = \{4\}$ |
| Read 5 | Update 4 | $\mathcal{F}_4 = \{5\}$ |
| Read 6 | Update 5 | $\mathcal{F}_5 = \{6\}$ |
| | Update 6 | $\mathcal{F}_6 = \{\}$ |

**(C) Fixed delay of 2**

Left matrix:

| | $g_1$ | $g_2$ | $g_3$ | $g_4$ | $g_5$ | $g_6$ |
|---|---|---|---|---|---|---|
| $g_1$ | 1 | 2 | 3 | | | |
| $g_2$ | 2 | 2 | 3 | 4 | | |
| $g_3$ | 3 | 3 | 3 | 4 | 5 | |
| $g_4$ | | 4 | 4 | 4 | 5 | 6 |
| $g_5$ | | | 5 | 5 | 5 | 6 |
| $g_6$ | | | | 6 | 6 | 6 |

Right matrix:

| | $g_1$ | $g_2$ | $g_3$ | $g_4$ | $g_5$ | $g_6$ |
|---|---|---|---|---|---|---|
| $g_1$ | 1 | 1 | 1 | | | |
| $g_2$ | 1 | 2 | 2 | 2 | | |
| $g_3$ | 1 | 2 | 3 | 3 | 3 | |
| $g_4$ | | 2 | 3 | 4 | 4 | 4 |
| $g_5$ | | | 3 | 4 | 5 | 5 |
| $g_6$ | | | | 4 | 5 | 6 |

| | | |
|---|---|---|
| Read 1,2,3 | Update 1 | $\mathcal{F}_1 = \{2,3\}$ |
| Read 4 | Update 2 | $\mathcal{F}_2 = \{3,4\}$ |
| Read 5 | Update 3 | $\mathcal{F}_3 = \{4,5\}$ |
| Read 6 | Update 4 | $\mathcal{F}_4 = \{5,6\}$ |
| | Update 5 | $\mathcal{F}_5 = \{6\}$ |
| | Update 6 | $\mathcal{F}_6 = \{\}$ |

**(D) Arbitrary order**

Left matrix:

| | $g_1$ | $g_2$ | $g_3$ | $g_4$ | $g_5$ | $g_6$ |
|---|---|---|---|---|---|---|
| $g_1$ | 1 | | 3 | | | |
| $g_2$ | | 2 | 3 | | 5 | |
| $g_3$ | 3 | 3 | 3 | | 5 | |
| $g_4$ | | | | 4 | 5 | |
| $g_5$ | | 5 | 5 | 5 | 5 | 6 |
| $g_6$ | | | | | 6 | 6 |

Right matrix:

| | $g_1$ | $g_2$ | $g_3$ | $g_4$ | $g_5$ | $g_6$ |
|---|---|---|---|---|---|---|
| $g_1$ | 1 | | 1 | | | |
| $g_2$ | | 2 | 2 | | 2 | |
| $g_3$ | 1 | 2 | 3 | | 3 | |
| $g_4$ | | | | 4 | 4 | |
| $g_5$ | | 2 | 3 | 4 | 5 | 5 |
| $g_6$ | | | | | 5 | 6 |

| | | |
|---|---|---|
| Read 1, 3 | Update 1 | $\mathcal{F}_1 = \{3\}$ |
| Read 2, 5 | Update 2 | $\mathcal{F}_2 = \{3,5\}$ |
| | Update 3 | $\mathcal{F}_3 = \{5\}$ |
| Read 4 | Update 4 | $\mathcal{F}_4 = \{5\}$ |
| Read 6 | Update 5 | $\mathcal{F}_5 = \{6\}$ |
| | Update 6 | $\mathcal{F}_6 = \{\}$ |

Figure 3: Each row corresponds to a different delay pattern: batches of size 3, a constant delay of 2, a constant delay of 3, and an arbitrary delay pattern. Each pattern is shown as symmetric matrix, where cell $(i,j)$ with $i > j$ is gray if update $j$ is outstanding when update $i$ is applied. The left column emphasizes the backward gradient sums associated with each update: In particular, letting $B_t$ be the set of cells $(i,j)$ labeled $t$, we have $g_t(g_t + 2g_t^{\text{bck}}) = \sum_{(i,j)\in B_t} g_i g_j$. Similarly, for the right-hand matrices, letting $F_t$ be the set of cells $(i,j)$ labeled $t$, we have $g_t(g_t + 2g_t^{\text{fwd}}) = \sum_{(i,j)\in B_t} g_i g_j$. These quantities play a pivotal role in our algorithms and analysis.

# B  Comparing Regret Bounds for Sparse Learning with Delays

In this section, we compare regret bounds for a variety of algorithms under both a fully-adversarial assumption, and a stochastic sparsity assumption like the one used by Duchi et al. [11]. For simplicity, consider a single dimension, $T$ rounds, and gradients $g_t \in \{-1, 0, 1\}$. When appropriate, we consider a feasible set $[-R, R]$; we neglect the potential difference between $R$ and $\tilde{R}$ when comparing to our regret bounds, i.e., we assume $R = \tilde{R}$. In the fully adversarial model we assume exactly

|       | $g_1$ | $g_2$ | $g_3$ | $g_4$ | $g_5$ | $g_6$ |
|-------|-------|-------|-------|-------|-------|-------|
| $g_1$ | **1** | 1 | 1 |   |   |   |
| $g_2$ | 1 | **2** | 2 | 2 |   |   |
| $g_3$ | 1 | 2 | **3** | 3 | 3 |   |
| $g_4$ |   | 2 | 3 | **4** | 4 | 4 |
| $g_5$ |   |   | 3 | 4 | **5** | 5 |
| $g_6$ |   |   |   | 4 | 5 | **6** |

Read 1,2,3   r(1) = r(2) = r(3) = 1   Update 1   $\mathcal{F}_1 = \{2,3\}$
Read 4   r(4) = 2   Update 2   $\mathcal{F}_2 = \{3,4\}$
Read 5   r(5) = 3   Update 3   $\mathcal{F}_3 = \{4,5\}$
Read 6   r(6) = 4   Update 4   $\mathcal{F}_4 = \{5,6\}$
  Update 5   $\mathcal{F}_5 = \{6\}$
  Update 6   $\mathcal{F}_6 = \{\}$

Consider the difference $x_5 - \hat{x}_5$, where we have:

$$x_5 = -\beta_3 g_1 - \beta_4 g_2 - \beta_4 g_3 - \beta_4 g_4 \qquad \texttt{AdaptiveRevision}$$
$$\hat{x}_5 = -\beta_3 g_1 - \beta_4 g_2 - \beta_5 g_3 - \beta_6 g_4 \qquad \texttt{HypBack}$$

Figure 4: An example the difference between `HypBack` (which plays $\hat{x}_5$) and `AdaptiveRevision` (which plays $x_5$), in terms of the common learning rates $\beta_t$.

$(1-p)T$ of the gradients are 0, but the adversary chooses the gradients arbitrarily subject to this constraint. Under the stochastic sparsity assumption, each $g_t$ is exactly 0 with probability $1-p$ (chosen independently for each $t$); if $g_t$ is not zero, the adversary chooses it arbitrarily (WLOG from $\{-1, 1\}$).

**Synchronized minibatches**   We consider a mini-batch delay pattern with batches of size $m$. Of course, enforcing such a delay pattern requires synchronization: one of the key questions addressed here is whether similar bounds are possible with arbitrary delay patterns (with a maximum delay of $m$, say). We index batches by $j$, and let $b_j$ be the sum of the $m$ gradients in batch $j$. We assume $m$ divides $T$ so $J \equiv T/m$ is the total number of batches.

Without delays and given dense gradients $(p = 1)$, standard online gradient descent with an appropriate adaptive learning rate can achieve a bound of $R\sqrt{\sum_{t=1}^{T} g_t^2}$ (we generally ignore constants in this section).

Thus, in the adversarial delayed case, but with a minibatch delay pattern, we can run $J$ steps of online gradient descent on the combined gradients $b_j$, for a bound of $R\sqrt{\sum_{j=1}^{J} b_j^2} \leq R\sqrt{m^2 J} = \sqrt{mT}$ (when $p = 1$). This is the best we can do in the worst case. If only a $p$ fraction of the $g_t$ are non-zero, then we get a bound of $\sqrt{mpT}$, as the adversary can simply put all the non-zero gradients in the first $pT$ rounds.

If, instead, the non-zero rounds are chosen randomly with probability $p$, but the adversary still controls what the gradient is (given that it is nonzero), he can still ensures all non-zero gradients in the same batch are in the same direction. Then $b_j$ has binomial distribution (with an adversary-controlled sign), and so

$$\mathbb{E}[b_j^2] = \mathrm{Var}[b_j] + \mathrm{Mean}[b_j]^2 = mp(1-p) + (mp)^2 = mp(1 + mp - p).$$

Again, starting from the bound $R\sqrt{\sum_j b_j^2}$, taking expectations and applying Jensen's inequality, we have

$$\mathbb{E}\left[\sqrt{\sum_{j=1}^{J} b_j^2}\right] \leq \sqrt{\mathbb{E}\left[\sum_{j=1}^{J} b_j^2\right]} \leq \sqrt{Jmp(1+mp-p)} = \sqrt{(1+p(m-1))pT}.$$

Note that $1 + p(m-1) \leq m$, and so we have a strictly sharper bound than the $\sqrt{mpT}$ result when $p < 1$. Replacing $m - 1$ with $m$ does not weaken the bound in practice, and so we have

$$\text{Regret} \leq \sqrt{(1 + pm)pT}. \tag{10}$$

To see the improvement over the fully adversarial case, suppose $p = 1/100$ and $m = 100$. Then against an adversary we have regret $\sqrt{T}$, but with stochastic sparsity we have regret less than $\sqrt{T/50}$.

**Subsampling data**   Suppose we have $m$ machines. We should be able to just train on a $1/m$ fraction of the data on a single machine sequentially, and our regret will simply be about $m$ times the regret this machine sees. This gives us $m\sqrt{T/m} = \sqrt{mT}$ in the dense case. So in the worst case we get the same bound by subsampling that we got by doing minibatches of size $m$ (where we looked at $m$ times as much data and did $m$ times as much work!). Intuitively, this is because we don't account for any reduction in the variance of the gradient estimate due to increasing the minibatch size (because in a worst-case world that may not happen). Similarly, we get the same $\sqrt{mpT}$ bound when data is sparse but the adversary controls the sparsity.

However, in the stochastic sparsity model, the subsampling approach still gets a bound of only of $\sqrt{mpT}$, so now minibatching has an advantage, obtaining the better bound of (10).

**AdaptiveRevision**   The `AdaptiveRevision` algorithm achieves a bound like $\sqrt{mpT}$ in the fully-adversarial case, and a bound like (10) if we run the algorithm on a problem with stochastic sparsity, without knowing the delay pattern or sparsity in advance. Recall we have bounds of the form

$$\sqrt{\sum_{t=1}^{T} g_t(g_t + g_t^{\text{fwd}})} = \sqrt{G_{1:t}^{\text{fwd}}}.$$

First, in the fully-adversarial setting, again we can assume all the non-zeros occur in the first $pT$ rounds. On these rounds each $g_t$ is bounded by 1 and $g_t^{\text{fwd}}$ is bounded by $m$, and so we immediately have a bound of $\sqrt{mpT}$.

Now we consider a stochastic sparsity pattern. Under the assumptions of this section, note $\mathbb{E}[g_t^2] = p$ and $\mathbb{E}[|g_t^{\text{fwd}}|] \leq mp$, and so, taking advantage of independence,

$$\mathbb{E}[g_t(g_t + g_t^{\text{fwd}})] \leq \mathbb{E}[g_t^2] + \mathbb{E}[|g_t|]\,\mathbb{E}[|g_t^{\text{fwd}}|] = p + mp^2.$$

Thus, taking expectations, we have

$$\mathbb{E}\left[\sqrt{\sum_t G_{1:t}^{\text{fwd}}}\right] \leq \sqrt{\mathbb{E}\left[\sum_t G_{1:t}^{\text{fwd}}\right]} \leq \sqrt{(1+mp)pT},$$

matching the performance of the synchronized mini-batch algorithm, (10). Note, however, we achieve this bound asynchronously, for arbitrary delay patterns, and without needing to know $m$ or $p$ in advance.

**AsyncDA and AsyncAdaGrad**   We now compare these results to those of Duchi et al. [11]. Under our assumptions, we have (using their notation for the moment) $\|x^*\|_2 = 1$, $M_j = 1$, and $M = 1$, and consider a single coordinate $j$. Then, with $m = 1$, their lower bound matches the $\sqrt{pT}$ result above, and this is achieved by (synchronous) OGD or Dual Averaging (their Eq. 5) and by by AdaGrad (their Eq. 6). Their results for AsyncDA and AsyncAdaGrad apply in the stochastic sparsity model.

For AsyncDA, note that in our simple case $\mathsf{M}^2 = \mathbb{E}[g_t^2] \leq p \cdot 1 + (1-p) \cdot 0 = p$ and $M_j = 1$. Then, their Theorem 3 becomes

$$\mathbb{E}[\text{Regret}] \leq \frac{1}{2\eta} + \frac{\eta}{2}Tp + \eta Tmp^2.$$

With an optimal learning rate $\eta = 1/\sqrt{T + 2mp^2T}$ chosen with knowledge of both $m$ and $p$, this gives regret $\sqrt{(1 + 2pm)pT}$, which matches (10). On the other hand, generally $p$ cannot be known in advance in an online setting, so using $\eta = 1/\sqrt{T + 2mT}$ gives only $\mathcal{O}\big((1 + p^2)\sqrt{mT}\big)$. This bound is significantly worse as $p \to 0$.

For AsyncAdaGrad, their Theorem 5 becomes

$$\mathbb{E}[\text{Regret}] \leq \frac{1}{\eta}\sqrt{m + Tp} + \eta\sqrt{Tp}(1 + pm).$$

With a learning rate scale factor of $\eta = \frac{1}{\sqrt{1+mp}}$ (again, dependent on both $m$ and $p$), this gives a bond that is $\mathcal{O}\big(\sqrt{(1+pm)(m+pT)}\big)$, which matches (10) when we ignore terms independent of

$T$ (noting $\sqrt{m+pT} \le \sqrt{m} + \sqrt{pT}$). Without knowledge of $p$ (say, taking $\eta = \frac{1}{\sqrt{1+m}}$), we arrive at bound like $\mathcal{O}\big((1+p)\sqrt{mpT}\big)$; without knowledge of $m$ or $p$, we arrive at a bound no better than $\mathcal{O}\big((1+pm)\sqrt{Tp}\big)$ (e.g., taking $\eta = 1$).

## C   Complete Analysis and Proofs

Several results will depend on the following basic result:

**Lemma 4.** *Under the above definitions, we have*

A.  $t \in \mathcal{F}_{r(t)} \iff t \ne r(t)$

B.  $o(t) \ge t$

C.  $s \le t \Rightarrow o(s) \le o(t)$

D.  $o(r(t)) \ge t$

E.  $t \in \mathcal{B}_s \iff s \in \mathcal{F}_t$

F.  `InOrder` *implies* $s_1 \le s_2 \Rightarrow r(s_1) \le r(s_2)$

G.  *If delay is bounded by* $m$, *then* $o(t) \le t+m$.

It is worth remarking that our choice of indices ensures $g_t^{\mathrm{bck}}$ is a sum of consecutively-indexed updates, while this need not be the case for $g_t^{\mathrm{fwd}}$. However, the `InOrder` property in fact implies $G^{\mathrm{fwd}}$ is sum of consecutively indexed gradients.

*Proof.* Most of these are immediate consequences of the definitions. For claim (C), first note if $s = t$, we are done. Suppose $s < t$, and consider two cases. First, suppose $o(s) \le t$, then $o(s) \le o(t)$ using (B), and we are done. For the second case, suppose $o(s) > t$. Then since $s < t$, we have $o(s) \in \mathcal{F}_t$, implying $o(t) \ge o(s)$. For (D), if $r(t) = t$, we are done by (B). If $r(t) < t$, then $t \in \mathcal{F}_{r(t)}$ (A), and so $o(r(t)) \ge t$. For (E), suppose $t \in \mathcal{B}_s = \{r(s), \dots, s-1\}$, so $r(s) \le t < s$, and so $t \in \mathcal{F}_s$. For the other direction, if $s \in \mathcal{F}_t$, we have $r(s) \le t < s$, which implies $t \in \{r(s), \dots, s-1\} = \mathcal{B}_s$. Claim (F) is the contrapositive of the definition of `InOrder`. For (G), if $o(t) = t$, we are done. Otherwise, let $s = o(t)$ with $s \in \mathcal{F}_t$, and so $r(s) \le t < s$. Then, $o(t) - t = s - t \le s - r(s) \le m$. $\qquad\square$

### C.1   Proof of Lemma 2

The analysis will use the following result:

**Lemma 5.** *Assume delays are bounded by* $m$ *and* $|g_t| \le L$. *Then, given* `InOrder` *delays, for all* $t$,

$$G_{1:t}^{\mathrm{fwd}} - m^2 L^2 \le G_{1:o(t)}^{\mathrm{bck}} \le G_{1:t}^{\mathrm{fwd}} + m^2 L^2.$$

*Proof.* Note

$$G_{1:o(t)}^{\mathrm{bck}} = \sum_{u=1}^{o(t)} g_u^2 + 2\sum_{u=1}^{o(t)} g_u \sum_{s \in \mathcal{B}_u} g_s.$$

Considering the last term,

$$
\begin{aligned}
\sum_{u=1}^{o(t)} g_u \sum_{s \in \mathcal{B}_u} g_s &= \sum_{u=1}^{o(t)}\sum_{s=1}^{o(t)} I(s \in \mathcal{B}_u) g_u g_s \\
&= \sum_{s=1}^{o(t)}\sum_{u=1}^{o(t)} I(u \in \mathcal{F}_s) g_u g_s && \text{Lemma 4(E).} \\
&= \sum_{s=1}^{t}\sum_{u \in \mathcal{F}_s} g_u g_s + \sum_{s=t+1}^{o(t)}\sum_{u=s+1}^{o(t)} I(u \in \mathcal{F}_s) g_u g_s.
\end{aligned}
$$

For the first part of the sum, observe that since $s \le t$ we have $u \in \mathcal{F}_s \Rightarrow 1 \le u \le o(t)$; in the second part of the sum, we can start indexing at $u = s+1$ since $u \in \mathcal{F}_s \Rightarrow u > s$. Plugging back

in, and dividing the sum over $g_u^2$ between the two terms,

$$G_{1:o(t)}^{\mathrm{bck}} = G_{1:t}^{\mathrm{fwd}} + \sum_{s=t+1}^{o(t)} \left( g_s^2 + 2 \sum_{u=s+1}^{o(t)} I(u \in \mathcal{F}_s) g_u g_s \right).$$

The result follows by observing there are at most $m^2$ terms of the form $g_s^2$ or $g_u g_s$ in the right-hand sum, and each of these is bounded by $L^2$. □

*Proof of Lemma 2.* Applying Lemma 1, we have

$$\mathrm{Regret} \le \frac{2R_T^2}{\hat{\eta}_T} + \frac{1}{2} \sum_{t=1}^{T} \hat{\eta}_t G_t^{\mathrm{fwd}}$$

where the learning rates $\hat{\eta}_t$ are given by (8). Lemma 5 implies $\tilde{G}_{1:o(t)}^{\mathrm{bck}} + m^2 L^2 \ge \tilde{G}_{1:t}^{\mathrm{fwd}}$, which in turn implies $\hat{\eta}_t \le \eta_t$. Thus,

$$\frac{1}{2} \sum_{t=1}^{T} \hat{\eta}_t G_t^{\mathrm{fwd}} \le \frac{1}{2} \sum_{t=1}^{T} \eta_t G_t^{\mathrm{fwd}} \le \alpha \sqrt{\tilde{G}_{1:T}^{\mathrm{fwd}}},$$

where we have again used Corollary 10. However, we could have

$$\tilde{G}_{1:T}^{\mathrm{bck}} > \tilde{G}_{1:T}^{\mathrm{fwd}}$$

(even though $G_{1:T}^{\mathrm{bck}} = G_{1:T}^{\mathrm{fwd}}$), but we can still bound the second term as

$$\frac{2R_T^2}{\hat{\eta}_T} = \frac{2R_T^2}{\alpha} \sqrt{\tilde{G}_{1:T}^{\mathrm{bck}} + G_0} \le \sqrt{2} \tilde{R} \sqrt{\tilde{G}_{1:T}^{\mathrm{bck}} + m^2 L^2} \le \sqrt{2} \tilde{R} \sqrt{\tilde{G}_{1:T}^{\mathrm{fwd}} + 2m^2 L^2},$$

using Lemma 5, $\alpha = \sqrt{2}\tilde{R}$, and $G_0 = m^2 L^2$. Recalling $\sqrt{a+b} \le \sqrt{a} + \sqrt{b}$ for $a, b \ge 0$ and then combining these results completes the proof. □

## C.2 Lemma 6

**Lemma 6.** *Under the* `InOrder` *assumption, Algorithm 1 plays the points specified by* (9).

The proof is a straightforward is straightforward induction making use of the following simpler expression for the points played by `AdaptiveRevision`:

**Lemma 7.** *Under the* `InOrder` *assumption, an equivalent expression for* (9) *(for $t \ge 2$) is*

$$x_t = - \sum_{s=1}^{r(t)-1} \beta_{o(s)} g_s - \sum_{s=r(t)}^{t-1} \beta_{t-1} g_s.$$

*Proof.* Starting from (9), it is sufficient to show the $\eta_s^t$ take on the claimed values. First, consider an $s < r(t)$. Note $r(o(s)) \le s$ since $o(s) \in \mathcal{F}_s \cup \{s\}$. Thus $r(o(s)) \le s < r(t)$, and so under `InOrder`, we have $o(s) < t$, which implies $o(s) \le t - 1$, and so $\eta_s^t = \beta_{o(s)}$.

Now suppose $s \ge r(t)$. Then $o(s) \ge o(r(t))$ and then $o(r(t)) \ge t$, using Lemma 4, parts (C) and (D), and so $\min(t-1, o(s)) = t - 1$, and so $\eta_s^t = \beta_{t-1}$. □

## C.3 Lemma 8

Let $\hat{x}_t$ be points played by `HypBack`, as in (8), and let $x_t$ be the points played by `AdaptiveRevision`. Then, we need to bound $\sum_{t=1}^{T} g_t(x_{r(t)} - \hat{x}_{r(t)})$, the difference in the loss incurred by `AdaptiveRevision` and `HypBack`. Figure 4 gives an example. The following lemma provides the needed guarantee. Note the gap is *independent* of the number of rounds $T$:

**Lemma 8.** *When* `InOrder` *holds, the maximum delay is $m$, and we take $G_0 = m^2 L^2$, we have* $\sum_{t=1}^{T} g_t(x_{r(t)} - \hat{x}_{r(t)}) \le 2\alpha L m$.

*Proof.* We begin by bounding

$$x_t - \hat{x}_t = \sum_{s=r(t)}^{t-1} -g_s\left(\eta_s^t - \hat{\eta}_s\right) = \sum_{s \in \mathcal{B}_t} -g_s\left(\beta_{t-1} - \beta_{o(s)}\right),$$

where we have used Lemma 7 and (8). For $1 \le s \le t$ and $d \ge 0$ define

$$\delta(s,t) \equiv \beta_s - \beta_t \qquad \text{and} \qquad \delta'(t,d) \equiv \beta_t - \beta_{t+d}.$$

Note $\delta$ and $\delta'$ are both decreasing in the first argument, and increasing in the second argument. When $r(t) = 1$ (for example, when $t = 1$), we have $\mathcal{B}_{r(t)} = \emptyset$, and so $x_{r(t)} - \hat{x}_{r(t)} = 0$; to handle this notationally, we let $\delta(0,t') = 0$ and $\delta'(0,d) = 0$ for any $t'$ and $d$. Then, we have

$$\sum_{t=1}^T g_t(x_{r(t)} - \hat{x}_{r(t)})$$

$$= -\sum_{t=1}^T g_t \sum_{s \in \mathcal{B}_{r(t)}} g_s \delta(r(t) - 1, o(s))$$

$$\le L^2 \sum_{t=1}^T \sum_{s \in \mathcal{B}_{r(t)}} \delta(r(t) - 1, o(s))$$

$$\le L^2 \sum_{t=1}^T m\delta(r(t) - 1, o(r(t) - 1))$$

where the last inequality uses Lemma 4(C) to show $\max_{s \in \mathcal{B}_{r(t)}} o(s) \le o(r(t) - 1)$, since $\max\left(\mathcal{B}_{r(t)}\right) = r(t) - 1$, and then notes $|\mathcal{B}_{r(t)}| \le m$. Continuing the inequality,

$$\le mL^2 \sum_{t=1}^T \delta(r(t) - 1, r(t) - 1 + m) \qquad \text{Lemma 4(G)}$$

$$= mL^2 \sum_{t=2}^T \delta'(r(t) - 1, m) \qquad \text{Defn., } \delta'(0, m) = 0$$

$$\le mL^2 \sum_{t=2}^T \delta'(\max(1, t - m - 1), m) \qquad \text{Since } r(t) \ge t - m.$$

We can bound the first $m$ terms by $m(mL^2)\frac{\alpha}{mL} = \alpha Lm$ since $\delta'(1, m) \le \beta_1 \le \frac{\alpha}{mL}$. Now, re-indexing the remaining terms, we have

$$mL^2 \sum_{t=1}^{T-m-1} \delta'(t, m) = mL^2 \sum_{t=1}^{T-m} (\beta_t - \beta_{t+m}) \le mL^2 \sum_{t=1}^m \beta_t \le \alpha mL,$$

where we have used the fact that this sum telescopes with an offset of $m$ terms, and we have again used $\beta_t \le \beta_1 \le \frac{\alpha}{mL}$. $\qquad \square$

## D  Technical Lemmas

We have the following slightly stronger version of the standard lemma (e.g., Auer et al. [21, Lemma 3.5]) used to analyze AdaGrad-style algorithms:

**Lemma 9.** *For any real numbers $x_1, x_2, \ldots, x_T$ such that $x_{1:t} > 0$ for $t \in \{1, \ldots, T\}$,*

$$\sum_{t=1}^T \frac{x_t}{\sqrt{x_{1:t}}} \le 2\sqrt{x_{1:T}}.$$

*Proof.* For $y \geq 0$, $\sqrt{y}$ is concave with derivative $\frac{1}{2\sqrt{y}}$, so by concavity for $z \geq 0$,

$$\sqrt{z} \leq \sqrt{y} + \frac{1}{2\sqrt{y}}(z - y).$$

For $a, b$ with $a \geq 0$ and $a + b \geq 0$, we can take $y = a + b$ and $z = a$, and so

$$2\sqrt{a} + \frac{b}{\sqrt{a+b}} \leq 2\sqrt{a+b}. \tag{11}$$

The proof proceeds by induction; the base case of $T = 1$ holds trivially, since $\sqrt{x_1} \leq 2\sqrt{x_1}$. Now, suppose the theorem holds for some $t \geq 1$. Then,

$$\sum_{s=1}^{t+1} \frac{x_s}{\sqrt{x_{1:s}}} = \sum_{s=1}^{t} \frac{x_s}{\sqrt{x_{1:s}}} + \frac{x_{t+1}}{\sqrt{x_{1:t+1}}}$$

$$\leq 2\sqrt{x_{1:t}} + \frac{x_{t+1}}{\sqrt{x_{1:t+1}}} \qquad \text{By the IH}$$

$$\leq 2\sqrt{x_{1:t+1}} \qquad \text{Using (11).}$$

$\square$

Using this, we can prove:

**Corollary 10.** *For any $x_1, x_2, \ldots, x_T \in \mathbb{R}$, with $x_1 > 0$, we have*

$$\sum_{t=1}^{T} \frac{x_t}{\sqrt{\max_{s \leq t} x_{1:s}}} \leq 2\sqrt{\max_{t \leq T} x_{1:t}}.$$

*Proof.* Define $z_t$ inductively with $z_1 = x_1 > 0$ such that $z_{1:t} = \max_{s \leq t} x_{1:t}$. Thus, $z_{1:t}$ is non-decreasing in $t$ so each $z_t \geq 0$. Thus, we can apply Lemma 9 to the sequence of $z_t$'s, which gives

$$\sum_{t=1}^{T} \frac{z_t}{\sqrt{z_{1:t}}} \leq 2\sqrt{z_{1:T}}.$$

To complete the proof, we argue by induction with the induction hypothesis

$$\sum_{t=1}^{T} \frac{z_t}{\sqrt{z_{1:t}}} - \sum_{t=1}^{T} \frac{x_t}{\sqrt{z_{1:t}}} \geq \frac{z_{1:T} - x_{1:T}}{\sqrt{z_{1:T}}}.$$

Observe the right-hand-side is non-negative by definition, so showing this is sufficient. The base case is trivial since $x_1 = z_1$. Suppose the IH holds for $T$. Then, we are adding

$$D = \frac{z_{T+1}}{\sqrt{z_{1:(T+1)}}} - \frac{x_{T+1}}{\sqrt{z_{1:(T+1)}}}$$

to the left-hand side. Further, since $z_{1:T} \leq z_{1:T+1}$, we have

$$\frac{z_{1:T} - x_{1:T}}{\sqrt{z_{1:T}}} \geq \frac{z_{1:T} - x_{1:T}}{\sqrt{z_{1:(T+1)}}}. \tag{12}$$

Thus,

$$\sum_{t=1}^{T+1} \frac{z_t}{\sqrt{z_{1:t}}} - \sum_{t=1}^{T+1} \frac{x_t}{\sqrt{z_{1:t}}} \geq \frac{z_{1:T} - x_{1:T}}{\sqrt{z_{1:T}}} + \frac{z_{T+1}}{\sqrt{z_{1:(T+1)}}} - \frac{x_{T+1}}{\sqrt{z_{1:(T+1)}}} \qquad \text{(IH)}$$

$$\geq \frac{z_{1:T+1} - x_{1:T+1}}{\sqrt{z_{1:T+1}}}. \qquad \text{Using (12)}$$

$\square$