[Reviews · NeurIPS 2014]

Submitted by Assigned_Reviewer_30

This paper considers asynchronous parallel updates in stochastic gradient descent with delays. This is a very important problem in large-scale distributed data processing. The objective of the problem studied in this paper is to achieve regret bounds similar to the ones obtained by adaptive gradient (i.e. AdaGrad) methods. This boils down to keeping track of updates to gradient coordinates. The paper considers three algorithms: a hypothetical (i.e. unimplementable) algorithm which uses information about the future updates to gradients to set the learning rate, and gives a AdaGrad style regret bound for it. The authors then approximate this algorithm with another hypothetical one, which cannot be implemented efficiently, which uses sums of backward gradient sums to set the learning rate, and prove that its regret is close to the first hypothetical algorithm. Finally, this second hypothetical algorithm is itself approximated by an efficient implementable algorithm, AdaptiveRevision, which has a similar regret bound.

Overall, the paper is quite well-written, and the sequence of reductions well explained. I believe that the results obtained are quite significant, and the problem is very well-motivated by practice. The solution given in this paper is quite nice.
Summary: This is a well-written paper which tackles the issue of delay gradient updates in stochastic gradient descent in a distributed environment, and gives an algorithm with a very nice analysis for it, obtaining regret bounds that are similar to the state-of-the-art AdaGrad algorithm. This is significant result for a difficult problem.

Submitted by Assigned_Reviewer_35

The paper proposes an asynchronous delay tolerant version of adaptive gradients for online learning in a distributed setting considering the effects of the delays. The main idea behind the adaptive gradients is to have different learning rates for different features where rarely updated features have higher learning rates. The paper proposes three algorithms to address the issue of delays in online learning, HypFwd and HypBack are impractical theoretical algorithms, while adaptive revision is an implementable version which approximates HypBack. The paper compares the performance of the proposed algorithms against two other algorithms AsyncAda-DA and AsyncAda-GD considering different delays patterns (constant, mini batch and random delay) on two real-life datasets. The empirical study showed that as the delays grew to 1000 or more, the proposed algorithms performed much better than the other methods.
Positives:
- The paper proposed a novel algorithm for online learning in a distributed manner which considers large update delays
- The studied problem is practically important
- The algorithm is theoretically characterized
- The experimental results show that the proposed algorithms work much better than the competing methods when delays are large
Negatives:
- In the experiments, the evaluation of the performance of the algorithms is done only for a single pass through the data. It would be interesting to see how the algorithm behaves if multiple passes are allowed
Summary: The paper proposes a theoretically justified and empirically successful novel online algorithm that can work with large delays and in an asynchronous mode. It is a nice contribution to the existing literature.

Submitted by Assigned_Reviewer_36

The authors proposed a delay-tolerant algorithm based on asynchronous distributed online learning method. The authors reordered the summation operation in the optimization process and achieved a more efficient asynchronous distributed optimization. the authors also proved a bound of the regret with the unprojected online gradient descent algorithm, while the bound seems not very tight.

experiments shows the proposed algorithm performs much better than other async algorithm with many threads. There are two concerns with the results.

One concern is the authors did not report the time comparison. As we know the main contribution of the paper is to replace the sync with asynchronous updating in the distributed environment. The overhead of sync is critical and major limitation of the distributed optimization. The authors proposed new algorithms which use async and achieve comparable performance with the sync algorithm which looks very promising, while the lack of time comparison, I can not tell how much improvement or contribution the algorithm can gain over existing algorithms. So the contribution of the paper is heavily discounted without this report.

Another concern is the authors seems to update each dimension separately, this is valuable in handling high dimensional data. While it does not state that the algorithm can also handle the long matrix optimization. In general the high dimensional problem is very rare, most of the cases we need to handle the big samples. Does the algorithm have any advantage over existing methods when dimension is low while sample size is huge? Also I suggest the authors perform more experiments to report the results in two cases.

Summary: The authors proposed new algorithm to use async instead of sync in distributed optimization and proved a loose regret bound. Experiments in several dataset seems the algorithm achieves better performance than other aync algorithmsd, but time is not reported.
Author Feedback
Author rebuttal: We thank the reviewers for their thoughtful reviews and constructive comments.

We agree with Reviewer 35 that evaluation in the batch setting (with multiple passes over the data) would be a good addition, and one we hope to make in a longer version of this work.

Reviewer 36 asked for a comparison of the time required by the synchronous and asynchronous algorithms. While we did not make this comparison explicitly, it's important to realize that even in the world of asynchronous algorithms, Adaptive Revision allows for a vastly larger degree of parallelism given a fixed desired accuracy. For example, the left two graphs of Figure 1 show that Adaptive Revision handles update delays about 20 times larger than those of competing algorithms with the same accuracy. All other things being equal, the update delay is proportional to the number of Readers operating in parallel. This means that, in a large-scale parallel setting, Adaptive Revision can optimize a function 20x faster than competing algorithms with the same accuracy.

Reviewer 36 also said that the high-dimensional problems considered in our experiments are "very rare", and suggested doing experiments with dense problems as well. While we agree that such a comparison could be interesting, high-dimensional problems are in fact quite common in industrial settings and are of great practical importance. Previous work cited, e.g. [11, 13, 14, 15, and 19], shows that many problems (e.g. ad click-through-rate prediction) are well modeled by high dimensional and sparse feature vectors, and further that AdaGrad-style algorithms perform particularly well on such representations. This is why we focused on the sparse case in our extension to the high-delay distributed setting.